# Harnessing synthetic lethality to predict the response to cancer treatment

Joo Sang Lee[1,2], Avinash Das[1], Livnat Jerby-Arnon[3], Rand Arafeh [4], Noam Auslander[1,2], Matthew Davidson [5], Lynn McGarry[5], Daniel James[5], Arnaud Amzallag[6,7,8], Seung Gu Park [1], Kuoyuan Cheng[1,2], Welles Robinson[1,2], Dikla Atias[9], Chani Stossel[9], Ella Buzhor[9], Gidi Stein[10], Joshua J. Waterfall [11], Paul S. Meltzer[11], Talia Golan[9,10], Sridhar Hannenhalli[1], Eyal Gottlieb[5], Cyril H. Benes[6,7], Yardena Samuels[4], Emma Shanks[5] & Eytan Ruppin[1,2,3,10]

While synthetic lethality (SL) holds promise in developing effective cancer therapies, SL candidates found via experimental screens often have limited translational value. Here we present a data-driven approach, ISLE (identification of clinically relevant synthetic lethality), that mines TCGA cohort to identify the most likely clinically relevant SL interactions (cSLi) from a given candidate set of lab-screened SLi. We first validate ISLE via a benchmark of large-scale drug response screens and by predicting drug efficacy in mouse xenograft models. We then experimentally test a select set of predicted cSLi via new screening experiments, validating their predicted context-specific sensitivity in hypoxic vs normoxic conditions and demonstrating cSLi's utility in predicting synergistic drug combinations. We show that cSLi can successfully predict patients' drug treatment response and provide patient stratification signatures. ISLE thus complements existing actionable mutation-based methods for precision cancer therapy, offering an opportunity to expand its scope to the whole genome.

[1] Center for Bioinformatics and Computational Biology, University of Maryland Institute of Advanced Computer Science (UMIACS) & Department of Computer Science, University of Maryland, College Park, MD 20742, USA. [2] Cancer Data Science Lab, National Cancer Institute, National Institute of Health, Bethesda, MD 20892, USA. [3] The Blavatnik School of Computer Science, Tel Aviv University, Tel Aviv 6997801, Israel. [4] Department of Molecular Cell Biology, Weizmann Institute, Rehovot 7610001, Israel. [5] Cancer Research UK, Beatson Institute, Switchback Road, Glasgow G61 1BD Scotland, UK. [6] Massachusetts General Hospital Center for Cancer Research, Charlestown, MA 02129, USA. [7] Harvard Medical School, Boston, MA 02114, USA. [8] PatientsLikeMe, 160 Second Street, Cambridge, MA 02142, USA. [9] Division of Oncology, Sheba Medical Center Tel Hashomer, Ramat-Gan 5262100, Israel. [10] The Sackler School of Medicine, Tel Aviv University, Tel Aviv 6997801, Israel. [11] Genetics Branch, National Cancer Institute, National Institutes of Health, Bethesda, MD 20892, USA. These authors contributed equally: Joo Sang Lee, Avinash Das. Correspondence and requests for materials should be addressed to E.R. (email: eytan.ruppin@nih.gov)

The success of precision oncology depends on its ability to translate accumulating genomic data into actionable treatment options tailored for individual patients. This requires identifying a genomic signature from patient tumor samples, then matching it with the most effective therapeutic options. With the ever-increasing volume of genomic data, the bottleneck now lies on how to extract patient-specific vulnerabilities from the data and connect them to patient prognosis and drug response. One promising way to tackle this challenge is based on the concept of synthetic lethality (SL). SL describes the relationship between two genes whereby an individual inactivation of either gene results in a viable phenotype, while their combined inactivation is lethal[1,2]. SL has long been considered a foundation for the development of selective anticancer therapies[1,3–5], which aim to inhibit the SL partner of a gene that is inactivated de novo in the cancer cells. Beyond guiding the development of novel selective cancer therapies, it has been noted that the network of SL interactions can provide a bird's eye view on the genomic state of a given tumor that can be leveraged to identify tumor-specific vulnerabilities and develop effective synergistic drug combination therapies in a precision-based manner[6,7].

Given the importance of SL, considerable work has been devoted to identifying such interactions in cancer—both experimentally[2,5] and computationally[8]. Experimentally, extensive efforts have been made to tease out the wiring of genetic interactions in cancer cells based on single cell lines[9–25] or large-scale knockout screens[26–30]. Computationally, various machine learning methodologies have been applied to predict genetic interactions in different species[8,31–35] and cancer (by utilizing yeast SL)[22,36], utilizing metabolic modeling[37,38], evolutionary characteristics[22,35], transcriptomic profiles[39–41], and more recently cancer patient data[42–46]. Nevertheless, so far the utility of SL in the clinic has been primarily limited to SLi in DNA damage pathways[47], and as we show further below, many of the SLs identified in current screens manifest a poor predictive signal in actual patients' data.

Here we present a statistical approach for identifying clinically relevant SL interaction (SLi) in a genome-scale manner, termed identification of clinically relevant synthetic lethality (ISLE). ISLE takes lab-screened SL interactions as inputs and analyzes tumor molecular profiles, patient clinical data, and gene phylogeny relations to identify SLi that are predictive of patients' drug response. The ISLE-identified SL interactions are shown to predict drug response to a wide variety of drugs both in vitro and in vivo, providing a basis for rational design of synergistic drug combinations. We first benchmark ISLE with large-scale in vitro drug response screens and in mouse xenograft models. We then experimentally test and validate predictions involving context-specific gene essentiality and drug efficacy, and synergistic drug combinations in patient-derived cell lines. We finally show that cSLi, which are inferred from mining untreated patients' data, can successfully predict treatment outcomes in cancer patients without any need for training on specific patient cohorts of drug response data. Taken together, these results offer a novel approach for precision-based cancer therapy from the patients' tumor data.

## Results

### Identifying clinically relevant SL interactions via ISLE. ISLE takes the initial pool of lab-identified candidate SL pairs as input and identifies among those the subset that is more likely to be clinically relevant, that is, supported by tumor data (see below). The initial pool is determined either by direct isogenic (or double-knockout) cell line screens (Initial Set I, Supplementary Data 1) or

guilt-by-association[23,45,48] using large-scale single gene knock-out experiments[26–30] (Initial Set II), creating a pool of total 16 million candidate SL pairs (see Methods, Supplementary Note 1). The two initial SL input sets show significant overlap, confirming findings of Wang et al.[23] (hypergeometric $P < 1.4E-28$) (see Supplementary Data 2). To identify putative cSLi from the initial pool; ISLE analyzes molecular, and survival data of patient tumor samples from The Cancer Genome Atlas (TCGA)[49] and evaluates the extent to which clinical data support in vitro screens. Conceptually, ISLE selects the clinically relevant SL pairs that satisfy all of the three conditions outlined below. From a computational standpoint, however, it applies them in a specific sequential manner that minimizes the computational cost of their identification (Fig. 1a, see Methods for full details):

First, ISLE mines gene expression and SCNA data of the input patient tumor samples to identify under-represented candidate gene pairs, whose co-inactivation is significantly less frequent than expected by their individual inactivation frequencies, testifying that their co-inactivation is under negative selection (using hypergeometric test; see Methods, Supplementary Note 1).

Second, ISLE selects SL pairs where the tumor samples with a given pair in a co-inactive state exhibit better patient's survival than the samples where it is not, testifying that this SL pair is likely to reduce tumor fitness when co-inactive. ISLE uses a stratified Cox proportional hazard model to establish this association while controlling for confounding factors including cancer type, genomic instability[50] and patients' gender, age, and ethnicity (see Methods).

Third, ISLE selects SL pairs composed of genes having high phylogenetic similarity, motivated by the observation that functionally interacting genes tend to co-evolve[1,22,51–53] (Supplementary Note 1). Those candidate pairs passing all steps compose the final output set of SL pairs predicted by ISLE (see Methods).

We applied ISLE to the initial input pool of candidate SL pairs identified above. ISLE analyzed SCNA, gene expression, mutation and survival data of 8749 clinical samples in TCGA across 28 cancer types[54], to identify a pan-cancer clinical-SL-network (Fig. 1a). The initial input set consisted of 16,375,526 candidate SLs, out of which 9.1% passed the first step, 0.4% of the initial pool pass the second step, and 0.1% of the initial pool pass the third step and constitute the final cSL network. The resulting cSL network is comprised of 8511 genes and 21,534 interactions (Supplementary Fig. 1, Supplementary Data 3, all networks are available online via an interactive interface, see Supplementary Note 2, the core SL network is presented in Fig. 1b, where a stricter FDR < 0.1 is used to obtain a smaller network for visualization purposes). The cSL genes are highly enriched in cell proliferation, migration, apoptosis, and signal transduction (FDR-corrected hypergeometric $P < 1E-8$, Supplementary Data 4). The identified cSL interactions are enriched with physical protein interactions (hypergeometric $P < 6.6E-3$)[55], consistent with previous observations in the literature[23,56]. The final cSL pairs successfully predict the survival response in an independent cohort of breast cancer patients[57], as expected (Supplementary Note 1).

Throughout the manuscript, we term the candidate SL pairs that withstand the three ISLE steps as clinically relevant SL (cSL), while those that are filtered out are termed non-clinically relevant SL (ncSL). Surprisingly, the ISLE analysis shows that only a small fraction of the gold standard initial pool (Initial Set I) is cSL: (1) only 12.5% of the pairs identified in the isogenic or double knockdown screens show evidence of negative selection in tumor samples (i.e., their co-downregulation is significantly less frequent than expected by chance, Step I). An even smaller selection of 0.2% of the original candidate pairs is associated with improved patient survival when co-downregulated (as would be expected

from cSLs, Step II). Importantly, to estimate the clinical relevance of these interactions, we used mutation data for isogeneic screens; and for double knockdown screens, we used copy number and transcriptomics data (see Supplementary Note 1 for details). Furthermore, in addition to pan-cancer tests, we also performed the clinical estimations outlined above in the specific cancer types of relevance, aiming to uncover the potential clinical relevance of the SLs screened in as comprehensive manner as possible (see Methods, Supplementary Note 1 for details). (2) A similar trend emerges when evaluating the clinical relevance of the SLi inferred via analyzing single knockdown screens (Initial Set II), with only 2.7% of the pairs from initial pools showing such evidence of negative selection (Step I) and 0.4% of pairs showing better prognosis (Step II), respectively (see Supplementary Data 5). Beyond these tests, our analysis shows that ncSL pairs are not predictive of drug response in cancer patients, and their predictive signal of in vitro drug response is markedly lesser than that of cSLs (see below).

**Validating ISLE-identified SL interactions.** We performed three validation steps to evaluate the capability of ISLE to identify clinically relevant SLi. First, to validate individual ISLE-inferred SL interactions, we analyzed an in vitro large-scale drug response screen[58] covering 24 drugs in 500 cancer cell lines. We tested whether a drug is more effective in cell lines where the ISLE-inferred cSL partner(s) of its targets are downregulated (according to the cell lines gene expression), thus confirming the existence of pertaining SLi (Methods). The cSLi inferred by ISLE are markedly more predictive (Area under the curve (AUC) = 0.75, Fig. 2a) than the ncSL pairs (AUCs of 0.47, and precision-recall analysis in Supplementary Fig. 2). ISLE's performance level is also markedly superior to that of DAISY (AUCs of 0.37), and to that obtained by randomly shuffled networks (empirical $P < 1E-3$).

Second, we tested whether ISLE-identified cSLi predict single and combination drug response in patient-derived mouse

xenograft (PDX) models. We analyzed 375 samples of mouse models of 15 cancer types, which were treated with 36 single drugs[59]. As the first step in this analysis, ISLE was applied to identify the relevant drug-cSL interactions; that is, to identify the SL partners of each of the given drug targets (Methods; this results in an extended set of interactions compared to those identified in the overall cancer cSL network as the space of hypotheses that needs to be corrected for is now obviously much smaller). Second, to make the drug response predictions for each drug, we define its cSL-score in each sample, by denoting the number of its predicted drug-cSL partners that are downregulated in that sample according to its gene expression data divided by the number of drug targets (Eq. (4) in Methods). As expected, responders show significantly higher cSL-scores than non-responders for five out of seven drugs that have sufficient tumor response data available (using Wilcoxon rank sum test; Fig. 2b, Methods). Furthermore, the cSLi network successfully predicts the progression-free survival in mouse models (defined as the time for the tumor size to grow to twice the size of the baseline, see Methods, Supplementary Note 1) (overall log rank $P < 2.90E-6$, $\Delta AUC = 0.15$), which remains significant after controlling for cancer types (Cox hazard ratio = 1.28 ($P < 3.82E-3$)). Neither DAISY nor ncSL partners are predictive of any of the drugs considered, and ISLE's performance is significantly superior to that of randomly assigned SL partners (empirical $P < 1E-3$).

Third, to benchmark the cSL-based drug response predictions, we analyzed the DREAM7 challenge data[60]. We focused on 15 of the 28 drugs that have specific targets and applied ISLE to infer their cSL partners (see Methods, Supplementary Note 1). Drug efficacy in a cell line is predicted via its cSL-score inferred from the cell-line's transcriptomic profiles (as described in the previous paragraph). The resulting ISLE-based prediction is comparable to the top 5 supervised algorithms (trained in this specific data) reported in DREAM7[60] (Fig. 2c (left columns)). To predict

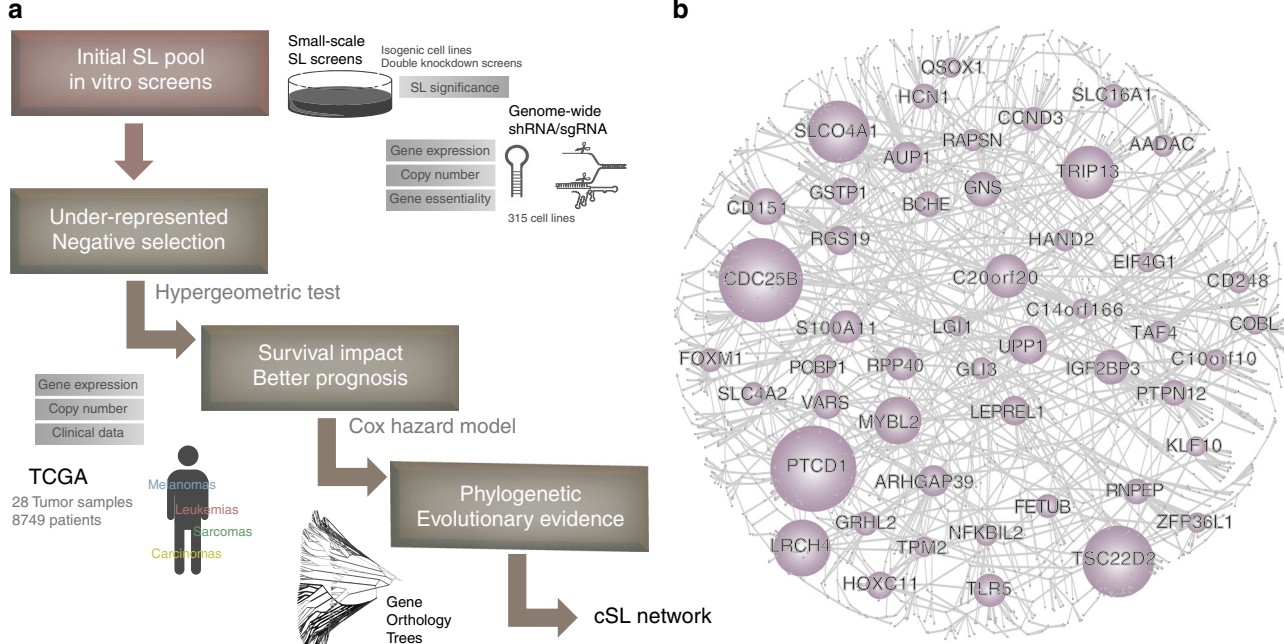

**Fig. 1** ISLE framework and the clinical-SL-network. **a** The three step inference procedure of ISLE and the datasets used in each step (Methods). **b** The core clinical-SL-network (with FDR < 0.1) includes 2326 interactions between 2153 genes, where the gene names having more than 10 cSL partners are marked; the size of nodes is proportional to the number of interactions they have). The complete network with FDR < 0.2 (the correction level used in all analyses presented in the paper) is provided in Supplementary Fig. 1 and in an interactive form at GitHub: https://github.com/jooslee/ISLE/

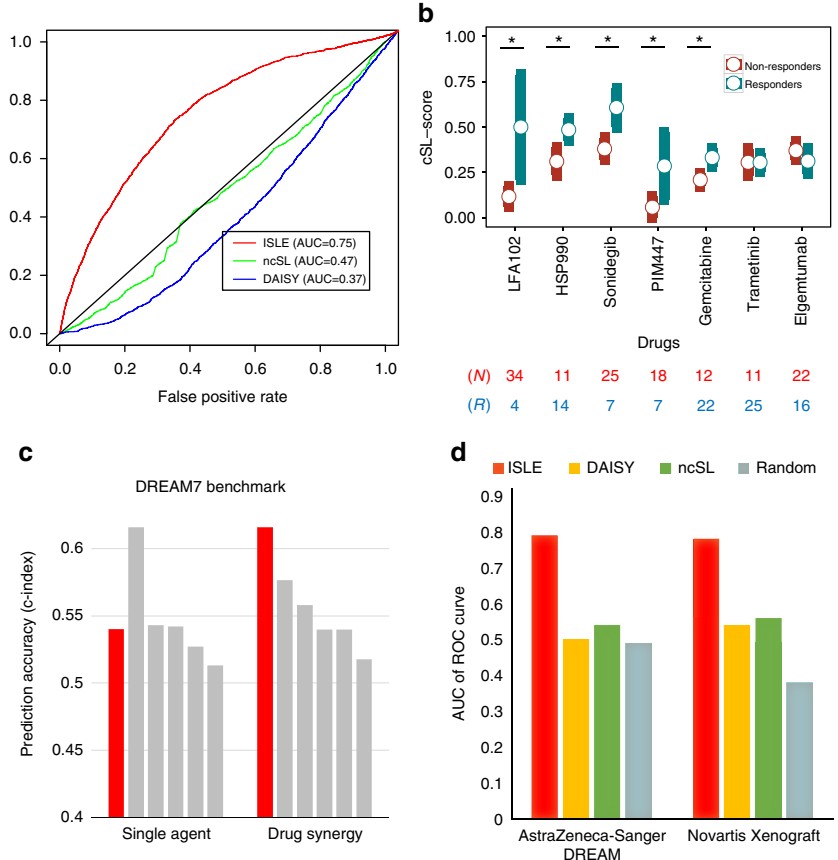

**Fig. 2** ISLE-based prediction of in vitro and in vivo drug response. **a** Prediction of in vitro drug response using drug-cSL-network in the CCLE[58] collections. The ROC curve compares the prediction performance of ISLE, ncSL, and DAISY (Methods). **b** Predicting in vivo drug response using drug-cSL-network. Mouse xenograft samples marked as responders (blue) show higher cSL-scores compared to the samples marked as non-responders (red). The X-axis shows seven drugs where sufficient drug response data are available and the Y-axis depicts the cSL-score (divided by the total number of SL partners to guide visualization, mean and s.e.m). (* marks the five drugs that are significantly predicted after multiple hypothesis correction (FDR-corrected Wilcoxon rank sum $P < 0.2$), and drugs are listed in order of significance). **c** Benchmarking ISLE-based drug response prediction versus the DREAM7 challenge. The figure shows prediction accuracy (evaluated using a variant of concordance index (Methods); Y-axis) of ISLE (red) and top five approaches (gray) both for the drug response to single agent (left columns) and drug combinations (right columns). **d** Predicting drug synergy. The AUC of ROC curves displaying the SL-based prediction accuracy of synergistic drug combination screens of a recent DREAM challenge, and a large collection of mouse xenograft models[59]. Results are shown for ISLE cSL interactions (red) and compared with the DAISY SL-network (yellow), ncSL network (green), and randomly permuted networks (gray)

double-drug response, we hypothesized that a pair of drugs will be synergistic if there exists a strongly predicted SL between their gene targets (Methods). We used the ISLE inferred SL strength between the drug targets (cSL-pair-score) as the predictor of synergism, focusing on 12 inhibitor compounds (66 combinations) excluding two non-specific drugs from the data. ISLE prediction accuracy ranks higher than the best five supervised algorithms reported in the DREAM7[61] (Fig. 2c (right columns)). Importantly, ISLE does not use any drug response data from the training set cell lines or post-treatment transcriptomic profiles that were used to train all the other competing approaches in the original challenge. This makes ISLE less prone to the risk of overfitting and broadly applicable for drug response prediction without the need of further training. The performance of randomly selected, DAISY-identified and most importantly, ncSL partners is markedly inferior to that of ISLE (Supplementary Fig. 3).

Fourth, we analyzed large-scale compound synergy screens from a more recent AstraZeneca-Sanger Drug Combination Prediction DREAM Challenge 2015[62], which covers a set of pairwise combinations of 69 drugs (169 drug combinations and 535 candidate gene pairs total) in 85 cancer cell lines. We predict synergistic drug combinations based on the working hypothesis that if there is a predicted cSLi between the gene targets of each of the two drugs, they are likely to be synergistic as described above. We labeled a drug pair as overall synergistic if it experimentally exhibits synergism across many cell lines (see Methods). The simple, unsupervised cSL-based predictor described above provides a fairly accurate prediction (AUC = 0.79, Fig. 2d, Methods), significantly higher than the random (empirical $P <$ 1E-3), DAISY-SL, or ncSL networks (Fig. 2d, Supplementary Fig. 4). We further tested whether the ISLE-identified cSL predict combinations' drug response in PDX models, which includes 375 samples of mouse models of 15 cancer types, which were treated with 26 double-drug combinations[59]. Our unsupervised cSL-based predictor classifies drug pairs as synergistic vs. non-synergistic (AUC = 0.78, Fig. 2d, Supplementary Fig. 5).

**Experimentally testing cSLi via phenotypic screens**. We performed three layers of experiments to test the performance of

ISLE in predicting gene essentiality and drug response for single agents and drug combinations.

First, we used ISLE-identified SL interactions to predict gene context-specific essentiality[6]. To this end, we conducted a large-scale shRNA knockdown (KD) screening in two different environmental conditions (21% $O_2$, "normoxia" and 0.1% $O_2$, "hypoxia") in a liv7k oral cancer cell line (Supplementary Data 6). As described in the previous section, we computed the cSL-score of each gene targeted, by identifying how many of its SL partners are downregulated in the condition studied (Methods). The computed scores are predictive of the individual KD effects on cellular growth (FDR-corrected $t$-test $P < 0.2$ between every bin, Fig. 3a), and of the context-specific differential KD effects in the two conditions (predicted correctly for 71% of the genes analyzed, Fig. 3b). Both context-generic and specific predictions were inferior when using either randomly selected, DAISY-identified, or ncSL partners (Supplementary Note 1).

Second, we examined the cSL network's ability to predict in vitro drug response. To this end, we performed two sets of experiments. (1) We treated liv7k oral cancer cell line studied above with 463 drugs (including non-cancer drugs) under hypoxia and normoxia conditions (Methods). We labeled the drugs within the top 33% observed growth inhibition in the experiments as effective and computed the drug-cSL-scores of their targets in the two specified conditions as before, but divided by the number of targets per each drug (Supplementary Data 7, Methods). Drugs with high cSL-scores show stronger drug response in each condition separately (AUCs of 0.75, AUPRC = 0.71, empirical $P < 1E-3$), and randomly permutated SL networks, DAISY, and ncSL network failed to obtain a predictive signal (Methods, Supplementary Fig. 6A). Analogous to the gene knockout experiment, the cSL-score successfully predicted in which condition (hypoxia vs normoxia) the drug would be more effective for 95% of the drugs tested (Spearman $R = 0.36$ ($P < 0.01$), Supplementary Note 1). (2) We additionally conducted a large-scale drug response screen covering 142 small molecule inhibitors in seven breast cancer cell lines of different subtypes (Supplementary Data 8). cSL-scores inferred from the transcriptomic profiles of these cell lines successfully predicted drug response (AUC = 0.73, AUPRC = 0.48, Supplementary Fig. 6B),

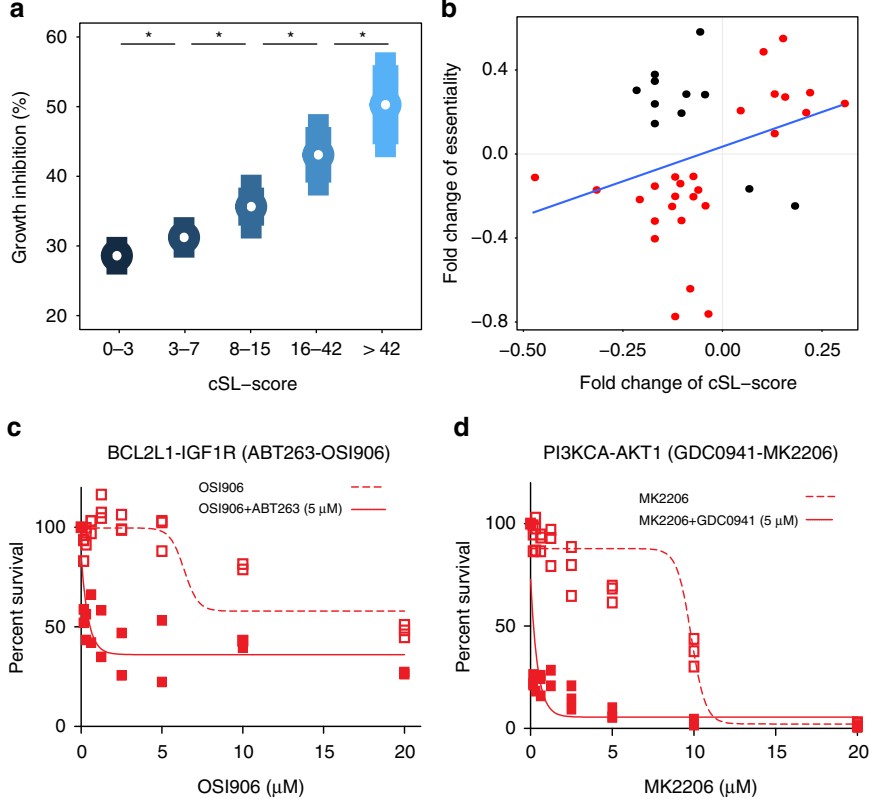

**Fig. 3** New experiments to test ISLE-based predictions on growth inhibition and drug combinations. **a, b** cSL-based prediction of growth inhibition in a knockdown screen in oral cancer. **a** Growth rate prediction: The number of downregulated cSL partners of a gene (X-axis; cSL-score) is associated with the percentage of growth inhibition observed after its knockdown (Y-axis; quantified as percent-growth inhibition compared to control, mean and s.e.m). Each bin of cSL-score shows significant differences (FDR-corrected $t$-test $P < 0.2^*$, see Supplementary Note 1). **b** cSL-based context-specific prediction of growth inhibition in hypoxic vs normoxic conditions. The fold change of cSL-score (X-axis in logscale) in normoxia vs. hypoxia shows a positive correlation with the corresponding growth inhibition fold change (Y-axis in logscale), correctly predicting the differentially observed growth inhibition in more than 71% of the 38 cases observed experimentally (marked as red dots in the 1st and 3rd quadrants). **c, d** The figures depict the representative dose response curves (see Supplementary Fig. 7B, C for other cell lines) of the predicted synergistic drug combinations of **c** ABT263 (BCL2L1 inhibitor) and the OSI906 (IGF1R inhibitor) and **d** GDC0941 (PIK3CA inhibitor) and the MK2206 (AKT1 inhibitor). The percentage of cell line survival (Y-axis) was measured at varying doses of OSI906 (respectively MK2206), with and without ABT263 (respectively GDC0941) treatment at 5uM (X-axis). The dashed lines denote the percentage of cell line survival at varying levels of OSI906 (MK2206) without ABT263 (GDC0941) treatments, and the solid lines denote the percentage of cell line survival at varying levels of OSI906 (MK2206) in the presence of 5 μM of ABT263 (GDC0941). The combined drug treatments are significantly more effective than the single treatments based on the analysis of variance ($P < 3.17E-11$ (**c**) and $P < 2.71E-8$ (**d**)). The Fa–CI curve for all drug combinations are in Supplementary Fig. 7D, and the full experimental measurements are presented in Supplementary Data 11, 12

**Table 1 SL partners and corresponding drug combinations tested in patient-derived melanoma cell lines**

| SL interaction | Drug combinations | Cell line 1 | Cell line 2 | Cell line 3 | Cell line 4 |
|---|---|---|---|---|---|
| BCL2L1-IGF1R | ABT263-OSI906 | 0.47 | 0.54 | 0.68 | 1.12 |
| PIK3CA-AKT1 | GDC0941-MK2206 | 0.75 | 0.23 | 0.77 | 0.59 |

The table shows the top two predicted cSL interactions tested in melanoma, the corresponding drugs tested and the experiments' outcome as evaluated by the combination index (CI) at 50% of cells are affected (CI < 0.7: strong synergism (blue), CI < 1: moderate synergism (green), and CI > 1: no synergism (white)).

while randomly selected partners are not predictive. DAISY-SL or ncSL partners are also not predictive for this task (AUC = 0.50 and 0.51, respectively in Supplementary Fig. 6).

Third, we conducted new drug combination experiments in patient-derived melanoma cell lines. Focusing on key melanoma drivers[63], we selected the top target pairs among the strongest ISLE-predicted cSLi between them—AKT1 and PIK3CA, and BCL2L1 and IGF1R (Table 1, Supplementary Data 9). We performed the experiments in cell lines where each gene of a selected pair was highly expressed (Supplementary Data 10), successfully validating seven out of the eight predicted cases (Table 1, see examples in Fig. 3c, d; see Supplementary Note 1 for details, Supplementary Fig. 7D for full Fa–CI curves and Supplementary Data 11, 12 for measurement data).

**cSL-based prediction of drug response in patients**. We next inferred the drug-cSL-network of clinically approved cancer drugs (Methods, Supplementary Fig. 8) and tested its capacity to predict drug response in three different patients' datasets of different tumor types[64,65,67]. Following the procedure described before regarding in vitro drug prediction, for each drug we predict its response in each sample by computing its cSL-score; the cSL-score of a drug denotes the number of its target genes' SL partners that are downregulated in the specific sample divided by the number of the drug's target genes (this number is determined from the sample's transcriptomics data; Eq. (4) in Methods). The first dataset is composed of 508 tumor biopsies of HER2-negative invasive breast cancer patients before treatment with taxane-anthracycline chemotherapy[64]. Patients with high cSL-scores have significantly longer distant relapse-free survival (DRFS) rates compared to those predicted as non-responders, as expected (Cox hazard ratio 1.40 ($P < 6.0E$-3), log rank $P$ of 9.2E-4, Fig. 4a). Indeed, patients annotated as responders show significantly higher cSL-scores than non-responders (Wilcoxon rank sum $P < 0.028$ with breast cancer subtypes controlled). The cSL partners of other drugs with the same degree distribution had no predictive signal, and so do ncSLs and DAISY based predictions (log rank $P > 0.37$).

Second, we applied the network to predict the response of 25 patients with recurrent or metastatic non-small cell lung cancer (NSCLC) to the EGFR-inhibitor erlotinib[65,66]. All patients had EGFR wild-type tumors. The number of downregulated EGFR-cSL partners is a marker of better prognosis (Fig. 4b) and shows significant association with patient survival (Cox hazard ratio = 2.15 ($P < 6.5E$-3), Supplementary Fig. 9A, B), which is significantly better than the predictive performance of the randomly permuted networks (empirical $P < 0.05$). The cEGFR-cSL partners predict specific response to erlotinib, as opposed to merely predicting patient survival: It failed to predict patients' response

in an independent arm of the same trial in which 37 NSCLC patients were treated with sorafenib, a VEGFR inhibitor (log rank $P > 0.95$, Supplementary Fig. 9C, D). The performance of the ISLE is superior in predicting disease progression after eight weeks of treatment ($t$-test $P = 1.5E$-3) compared to the original EMT signature ($t$-test $P = 0.052$)[67], and on par with other supervised clinical drug response predictors[68] (see Supplementary Note 1). Finally, DAISY or ncSL networks failed to predict the clinical response to erlotinib (log rank $P > 0.56$).

Third, we analyzed an International Cancer Genome Consortium (ICGC) cohort of 80 ovarian cancer patients treated with taxane-cisplatin chemotherapy[67]. The corresponding drug-cSL partners are underexpressed in responders compared to non-responders, as expected (Wilcoxon rank sum $P < 9.1E$-3, Fig. 4c). Reassuringly, the ISLE-based prediction is significantly better than random or shuffled SL partners (empirical $P < 0.04$ and 0.03, respectively); DAISY-SL or ncSL partners were not predictive either (Wilcoxon rank sum $P > 0.17$).

Finally, we predicted drug response in the TCGA compendium[54]. We considered six drugs that have response evaluation criteria in solid tumors (RECIST) information following treatment for at least 12 patients (for available cancer types). Four out of these six drugs show significantly higher cSL-scores in responders' tumors (blue bars) than in those of non-responders (red bars) (Fig. 4d, empirical $P < 0.05$). (Notably, the cSL network was inferred only from the samples that have no drug treatment record ($N = 6268$) so that the testing is performed strictly on unseen samples, Methods.) The predictive signal is completely lost when using random, shuffled, DAISY or ncSL partners (both experimentally identified and inferred).

**Discussion**

Many SL lab screens have led to the discovery of numerous important leads for further in vivo follow-up. However, our analysis revealed that only a small fraction of the SLs candidates emerging from such in vitro screens analysis show an evidence of negative selection or effect on patients' prognosis in the TCGA cohort (Supplementary Note 1). This realization has led us to develop ISLE, a method for identifying the subset of these in vitro SL pairs that are more likely to be clinically relevant. As we have shown throughout the paper, by and large cSL pairs enable successful prediction of response to a wide panel of targeted therapies in cancer patients, which is markedly superior to the predictive power of ncSL pairs.

The clinical significance estimation of the initial pool of candidate SLs should be distinguished between two different cases: (1) First, in the case of the specific candidate SLs that arise from isogenic (or double-knockout) cell lines screens, we specifically performed their clinical significance estimation by considering

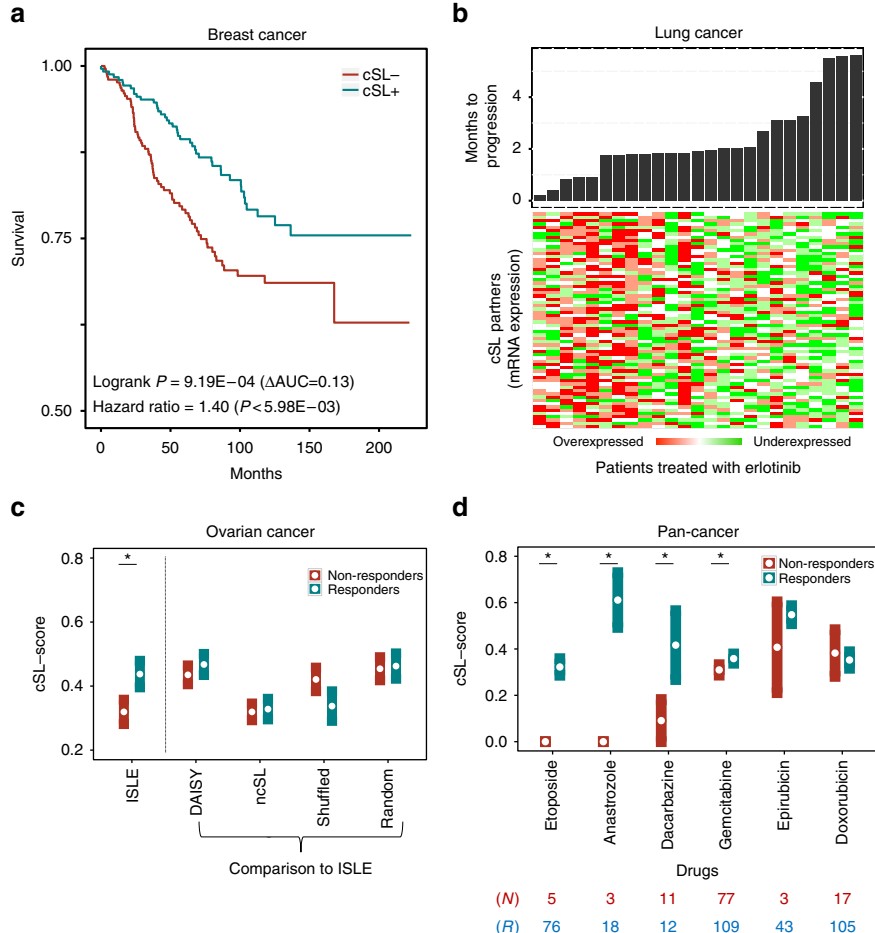

**Fig. 4** Drug-cSL-network predicts treatment outcomes in cancer patients. **a** The KM plot of predicted responders (blue) vs non-responders (red) to taxane-anthracycline chemotherapy[64]. We divided the patients into responders vs non-responders based on the median value of their cSL-scores. **b** The gene expression of the cSL partners in patients treated with erlotinib[65,66], ordered according to their months-to-progression (on-top). As predicted, patients with many downregulated cSL partners progressed slower. **c** The responders (blue) to taxane-cisplatin therapy for ovarian cancer show significantly higher ISLE cSL-scores than the non-responders (red) (Wilcoxon rank sum $P < 9.1E-3$). The $X$-axis shows the different groups of partners studied, and $Y$-axis provides their cSL-scores in responders vs non-responders. **d** TCGA patients with a large number of downregulated drug-cSL partners in their tumors show better response based on RECIST criteria. $X$-axis shows six drugs that have considerable (>12 samples) drug response information in TCGA, and $Y$-axis represents the cSL-score (divided by total number of SL partners to guide visualization, mean and s.e.m) of their drug targets, where cancer types are controlled for (* marks the four drugs that are significantly predicted after multiple hypothesis corrections (FDR-corrected Wilcoxon rank sum $P < 0.2$, and epirubicin FDR < 0.23), drugs are listed in order of significance). Blue (red) bars denote the cSL-scores of the responders (non-responders), and the numbers marked in blue (red) below the figure indicate the number of responders (non-responders) for each drug. All the analyses were performed using the drug-cSL-network (presented in Supplementary Fig. 8) based on the drug-target mapping available listed in Supplementary Data 13

the corresponding mutation (or copy number and expression) data in the tumors and also by scanning the specifically relevant cancer types in which they have been inferred, so we believe that their clinical significance estimations are quite robust. (2) Second, the clinical significance estimations of the candidate SLs emerging from the single-knockdown screens are however probably less tight, as they are both tested and inferred via a pan-cancer analysis (Methods), while some of these candidate SLi may still be of clinical value in specific cancer types and contexts (see Supplementary Note 1 for details).

The cSLi inferred are identified by analyzing a large cohort of untreated cancer samples, and then utilized to predict the response to cancer drugs without any further training, showing strong predictive power in a wide range of different data types (Supplementary Data 14). The absence of specific training on

post-treatment data reduces the risk of over-fitting and poor generalization and enables the prediction of response to new, untested drug candidates. ISLE's performance level is on par with numerous supervised prediction approaches employed for drug response prediction in the DREAM challenge (Fig. 2c). Importantly, it provides straightforward predictive cSLi signatures for each drug, which can be used to guide the selection of cell lines and models for further experimental screens, and for future patient stratification.

Like any other genome-wide computational prediction method, ISLE has several limitations that should be acknowledged: (1) first; the tumor data are noisy, both on the molecular and the survival side. To achieve a strong and robust predictive signal, we infer a common pan-cancer network by combining data from different tumor types, while obviously there is

variability in SLi at different cancer types. (2) Second, the current version of ISLE considers multitude types of evidence to increase the signal and controls for many potential confounders. However, additional ones could be further considered in the future when more data accumulates. (3) Third, by requiring that cSLi pass both molecular and survival filters we may miss true cSLs. Nevertheless, we prefer to take this more conservative approach that minimizes the number of cSLs that turn out not to be clinically relevant even if it incurs a higher rate of missed true interactions. Indeed, the results show that cSLi are markedly more predictive than non-cSLi, across the board.

Drugs targeting such genes that are high degree hubs in the cSL network may kill different sub-clones harboring different inactivated cSL partners and may reduce the likelihood of emerging resistance in heterogeneous tumors. With the accelerated accumulation of new patients' data, ISLE may be further improved by analyzing new types of omics data (e.g., more extensive sequence and epigenetic data) and lead to the generation of cancer type/context-specific SL networks. Taken together, ISLE complements existing mutation-based targeted approaches[69] by extending their scope to the whole genome.

## Methods

**Gene activity and FDR threshold settings**. Throughout the paper, we define the following unless specified otherwise: (A) A gene to be inactive in a sample if its gene expression (or SCNA) is below 1/3-quantile across samples in each cancer type. This is to account for the distinct basal expression levels of the gene in different cancer types (see Supplementary Note 1). We used both transcriptomic and SCNA profiles for identifying SLi using ISLE, but used only transcriptomic profiles for drug response prediction throughout the paper.

(B) A candidate SL pair to be significant when FDR-corrected $p$-value < 0.2 in identifying the cSL network. In the three steps of ISLE, the significant pairs were selected and passed to the next step. In all multiple hypothesis correction, we consistently used a uniform FDR threshold of 0.2.

**ISLE**. The first step incorporates the union of candidate SL pairs derived from two separate procedures. Initial Set I: the experimentally identified SL were collected from 17 screens, each performed in a single cell line[9–25]. The screens spanned altogether 154,707 potential pairwise interactions tested in eight different cancer types, among which 6033 pairs compose the positive set (about 4%, see Supplementary Data 1). We excluded from the screens those interactions that focus on activating driver mutation or copy number amplification, such as KRAS[14,17], PTTG1[17], and MYC[16], resulting in 4252 unique candidate SL pairs. Initial Set II: the SLi were inferred from five large-scale shRNA/sgRNA single gene knockout screens[26–30], spanning a total of 9,253,974 measurements in 315 cancer cell lines from 19 different tissues of origin, through 'guilt-by-association'[23,45,48]: By definition, it is expected that gene $A$ will be essential only when its SL partner gene $B$ is inactive in a given cancer cell line. Using a set of input genome-wide shRNA/sgRNA screens in a reference collection of cell lines, ISLE examines all current candidate SL pairs to identify those pairs that show conditional essentiality: Gene $A$ is defined as conditionally essential with gene $B$ if its essentiality is significantly higher in the samples where gene $B$ is underexpressed (defined above) using Wilcoxon rank sum test. SCNA based conditional essentiality is determined analogously. The candidate SL pairs that show significance either by mRNA or SCNA data selected in the set (see Supplementary Note 1). The union of Initial Set I and II has resulted in a total of 16,375,526 candidate SLi.

First, we identify under-represented SL pairs by analyzing tumor molecular data. The step searches for gene pairs whose co-inactivation are under negative selection. We mine gene expression and SCNA data of input tumor samples to identify gene pairs $A$ and $B$ whose co-inactivation (defined above) is significantly less frequent than expected (identified via a hypergeometric test). Formally, if $N$ is the total number of tumor samples, $n_A$ ($n_B$) is the number of samples with gene $A$ (gene $B$) underactive, respectively, and $n_{AB}$ is the number of samples with co-inactivation of $A$ and $B$, the significance of depletion was determined by hypergeometric($n_{AB}$, $n_A$, $N$, $n_B$) (see Supplementary Note 1). The depletion of the activity state using SCNA is inferred analogously.

Second, we identify survival-informative pairs by analyzing patient survival data. The step selects a gene pair $A$ and $B$ as SL if tumor samples with co-inactive A-B exhibit significantly better patient's survival than tumor samples without the co-inactivation. Specifically, ISLE uses the following stratified Cox proportional hazard model to check this association, while controlling for various confounding factors including the effect of respective genes, cancer types, genomic instability[50], sex, age, and race (shown here for expression analysis and a similar model is used

to analyze SCNA data):

$$h_g(t, \text{patient}) \sim h_{0g}(t) \exp\big(\beta_1 I(A, B) + \beta_2 g(A) + \beta_3 g(B) + \beta_4 \text{age} + \beta_5 \text{GII}\big), \quad (1)$$

where $g$ is an indicator variable over all possible combinations of patients' stratifications based on cancer type, race, and sex. $h_g$ is the hazard function (defined as the risk of death of patients per unit time), and $h_{0g}(t)$ is the baseline-hazard function at time $t$ of the $g$th stratification. The model contains four covariates: (i) $I$ ($A$, $B$): indicator variable if the SL is functionally active in the patient's tumor, (ii) $g$ ($A$) and (iii) $g(B)$: gene expression of $A$ and $B$, (iv) age: age of the patient, (v) GII: genomic instability index. GII measures the relative amplification or deletion of genes in a tumor based on the SCNA. Given $s_i$ be the absolute of log ratio of SCNA of gene $i$ in a sample relative to normal control, GII of the sample is given as in Bilal et al.[50]:

$$\text{GII} = 1/N \sum_{1}^{N} I(s_i > 1). \quad (2)$$

The $\beta$s are the regression coefficient parameters of the covariates, which quantify the effect of covariates on the survival. All covariates are normalized to $N$ (0,1). The $\beta$s are determined by standard likelihood maximization of the model[70] using the R-package "Survival". The significance of $\beta_1$, which is coefficient for the SL interaction term is determined by comparing the likelihood of the model with the null model without the interaction indicator $I(A, B)$ followed by a likelihood ratio test[70], i.e.,

$$h_{\text{null},g}(t, \text{patient}) \sim h_{0g}(t) \exp\big(\beta_2 g(A) + \beta_3 g(B) + \beta_4 \text{age} + \beta_5 \text{GII}\big). \quad (3)$$

The $p$-value obtained by the likelihood ratio test is corrected for multiple hypotheses. Pairs exhibiting the significant survival improvement either in gene expression or SCNA are passed on to the next screen.

Third, we identify phylogenetically linked SL pairs by analyzing phylogenetic profiles. SL pairs were found to be conserved across different species[1,22]. Based on this notion, we further filter and select SL pairs composed of genes having high phylogenetic similarity. This is done by comparing the phylogenetic profile of the two genes $A$ and $B$ in a candidate SL pair across 86 species (adopting the method of Tabach et al.[40,41]). We then calculate the phylogenetic similarity between $A$ and $B$ using a non-negative matrix factorization (NMF)[71], which measures the Euclidian distance while taking into account their phylogeny. To determine the threshold used for this step, we used the median phylogenetic similarity of the pairs in Initial Set I as it optimally separates the positive and negative sets among these pairs (see Supplementary Note 1).

**ISLE-based drug response prediction**. For drug response and essentiality predictions, we applied ISLE to a limited set of candidate SL pairs between the inactivated genes (either by drug or gene knockout) and all protein-coding genes, performing multiple hypothesis correction at each drug or gene level with a single uniform FDR threshold of 0.2 throughout the paper. The drugs were mapped to their targets primarily based on DrugBank[72], and we referred other sources such as CCLE[58], GDSC[73] and the literature with exception to target genes whose mechanism of action is explicitly denoted as an agonist in DrugBank (Supplementary Data 13). We excluded those drugs whose target information is not concrete from such databases and the literature. As an example, Supplementary Fig. 8 depicts the drug-cSL-network that was used for predicting drug response in patients, that covers 232 SLi, involving 14 drug targets (blue), their corresponding 16 drugs (green), and their 207 SL partners (red).

Throughout the manuscript, we made predictions for drug response to (1) single agents and (2) drug combinations using ISLE. We focused on inhibitory compounds with specific targets because only targeted inhibitors are relevant to SL. For single agents, we use the following procedure, which we term Procedure (1). We hypothesize that a drug will be more effective in cell lines/samples that have more underactive cSL partners of its targets. To this end, we (i) identified the cSL partners of the genes targeted by each of the drugs (using ISLE pipeline as described above), (ii) defined cSL-score for individual samples/cell lines (defined below), and (iii) used cSL-score to predict the drug response and evaluated the prediction accuracy by comparing it with experimentally/clinically measured drug response data.

$$\text{cSL-score}_j = \sum_{i=1}^{N_P} I(\text{mRNA}_{ij} < q_{ij})/N_T, \quad (4)$$

where cSL-score$_j$ denotes the cSL-score of the $j$th sample, mRNA$_{ij}$ is the pre-treatment gene expression of $i$th cSL partners in the $j$th sample; $q_i$ is the 1/3-quantile threshold of the cancer type to which the $j$th sample belongs, $I(x)$ is an indicator function whose value is 1 if $x$ is true and 0 otherwise. $N_P$ is the number of cSL partners of the given gene of interest, and $N_T$ is the number of the targets of a given drug.

For drug combinations, we use the following procedure, which we term Procedure (2). ISLE SL-network was applied to predict synergistic drug combinations based on the working hypothesis that a synergy between two compounds arises from an underlying SLi between the gene targets of each of the two drugs. Accordingly, we (i) applied ISLE to all possible pairs between the drug-target genes, (ii) predicted the compound synergy using the best cSL-pair-score (defined below) between their target genes, and (iii) evaluated the prediction accuracy versus experimentally measured synergy.

$$cSL\text{-}pair-score = r_{initial} + r_I + r_{II} + r_{III}, \qquad (5)$$

where cSL-pair-score is a qualitative measure combining the significance levels at the initial candidate SL pool and the subsequent three statistical tests, $r_{initial}$ denotes the significance based on the experimental in vitro screens. $r_I$, $r_{II}$, and $r_{III}$ denote the rank-normalized values (between 0 and 1, with 1 representing a pair with the highest significance and 0 with the lowest) of the statistical significance levels across all gene pairs tested for step I, II, and III, respectively.

We compared the performance of ISLE against control networks, including ncSL network (defined below), and random networks. (1) For gene essentiality prediction, the gene's SL partner was assigned randomly. (2) For drug response to single agents, the cSL partners of a drug was assigned either by (i) random genes or (ii) the randomly selected cSL partners of other drugs. (3) For drug response to combinations, the random networks were created by randomly shuffling the drug's identities. Also, we compared the performance of ISLE with that of DAISY[45] (see Supplementary Note 1).

To evaluate the importance of patient data in predicting drug response, we consider ncSL pairs, which are the SLi that belong to the initial pool of in vitro SL screens (see "Building an initial pool of candidate SL pairs" in Methods) but do not pass the three steps of ISLE. For gene essentiality or single drug response prediction, we compared the performance of ISLE cSL partners to that of ncSL partners by selecting the most significant $N$ ncSL partners (determined by the significance of in vitro screens) per drug, where $N$ is the number of the ISLE cSL partners. For drug synergism, we used the predictive performance of $r_{initial}$ (Eq. (5)) from in vitro screens like that of ncSL because the performance of each ISLE steps is independently evaluated.

**Validating ISLE-identified individual SL interactions.** To validate individual ISLE-inferred SLi, we analyzed a large-scale drug response screen (Cancer Cell Encyclopedia (CCLE)[58], covering IC50 values of a total of 24 drugs across 500 cancer cell lines with their corresponding gene expression). We focused on the inhibitor compounds whose response is highly variable across different cell lines (see Supplementary Note 1).

We test whether inactivation of cSL partners ($P$) of a drug target ($T$) is associated with better response to the drug (hence supporting the presence of SLi between $P$ and $T$). We thus used the cSL-pair-score of the $P$–$T$ pair (i.e. the features, as produced by ISLE pipeline) to predict the association between the low expression of $P$ and better response to the drug targeting $T$ (i.e. the labels, 1 if the low expression of P is associated with the better response to the drug targeting $T$, 0 otherwise) using Wilcoxon rank sum test with FDR < 0.2 and fold change >0.35 (see Supplementary Note 1). We evaluated the prediction accuracy of ISLE based on the standard ROC and precision-recall analysis and compared it to the performance of the control SL networks.

**Validating ISLE with in vivo drug response data.** For in vivo analysis, we analyzed the large-scale mouse xenograft dataset, which collects 36 single drug response screening of 375 mouse samples in 15 cancer types[59]. The drug response was marked based on in vivo response evaluation criteria in solid tumors for mouse (mRECIST) criteria, namely complete response (CR), partial response (PR), stable disease (SD), and progressive disease (PD) based on tumor size reduction over time after treatment compared to the baseline (see Methods in Gao et al.[59]). Following Procedure 1 (see 'ISLE-based drug response prediction' in Methods above) using the drug-cSL-network, we evaluated our prediction (cSL-score) vs. experimentally measured drug response. Based on the in vivo pathological drug response annotation, the samples were divided into responders (CR, PR, SD) vs non-responders (PD) and their cSL-score were compared using a Wilcoxon rank sum test. We focused on seven drugs that show sufficient variability in best average response (BestAvgResponse)[59] across different samples per drug (variance >10-percentile) and sufficient sample size for robust comparison (Supplementary Note 1). Also, we tested our prediction against in vivo Progression Free Survival (PFS) data, which measures the time for the tumor to grow double of the baseline tumor size (see Methods in Gao et al.[59]). We tested whether high cSL-score is associated with improved survival using Cox regression analysis while controlling for confounders such as cancer types.

**Benchmarking ISLE with DREAM7 challenge data.** For single drug response, we focused our analysis on a set of 20 inhibitor compounds that have highly specific targets ($N_{target} < 3$) and significant SL partners predicted by ISLE (with FDR threshold of 0.2). We followed Procedure 1 (see 'ISLE-based drug response prediction' in Methods above) using drug-cSL-network to make ISLE-based drug response prediction, and evaluated and compared the performance using weighted

probabilistic concordance index (wpc-index), that is the collective measure of concordance index for each drug, taking the variance of the response to individual drugs into account, as defined in Costello et al.[60].

For drug combination, we used the ISLE inferred SL significance (cSL-pair-score) between the drug targets as the predictor for synergism. We followed Procedure 2 (see 'ISLE-based drug response prediction' in Methods above) to make synergistic compound prediction, and evaluated and compared the performance using probabilistic concordance index (pc-index), taking the variance of the response to individual drugs into account, as defined in Bansal et al.[61]. (see Supplementary Note 1 for further details).

**Validating ISLE drug combination screens.** We followed Procedure 2 (see 'ISLE-based drug response prediction' in Methods above), and evaluated the prediction accuracy based on ROC analysis and prediction-recall statistics. We applied this approach to a recent DREAM challenge[62], which covers 169 drug combinations that involve 535 target gene pairs in 85 cancer cell lines (AstraZeneca-Sanger Drug Combination Prediction DREAM challenge 2015, different from the DREAM7 challenge. In this dataset, the single and double-drug response were measured across multiple dose conditions, and a synergism was defined by the difference between the actual combination response and the additive effect of the respective single agents. This was implemented based on Loewe model, summarized into a single synergy score[72,74,75] (see the Data Description of the AstraZeneca-Sanger Drug Combination Prediction DREAM challenge[62]). We labeled a drug pair to be synergistic based on its maximal synergy score.

For in vivo synergistic combination, we applied the same approach to a collection of in vivo mouse xenograft models that cover 26 drug combinations in 375 samples[59]. We define a drug pair is synergistic if it shows synergism in more than 2/3-quantile of the samples it was tested. In each sample, a drug pair is synergistic, if the double treatment effect is greater than the summation of respective single treatments. Specifically, we used the best drug response metric (BestAvgResponse), which measures the averaged value of the maximal tumor size reduction over multiple time points (see Methods in Gao et al.[59]) to quantify the effect of the double and single treatment.

**Testing cSL network with functional and drug screens.** We use liv7k cell line, an aggressive (T3N2b), HPV-negative cell line derived from a primary tumor of the tongue, in a patient who received no chemo/radiotherapy before tumor excision. Generously gifted by Professor Richard Shaw (Head and Neck Cancer Consultant, Liverpool, UK). Cells are maintained in keratinocyte SFM (Life Technologies), supplemented with 2 mM L-glutamine, 0.2 ng/ml epidermal growth factor (EGF) and 25 µg/ml bovine pituitary extract (BPE). The cell line was tested for mycoplasma contamination and authenticated by exome sequencing and CNV.

For drug repurposing screen in liv7k cell line, a panel of drugs comprised from FDA-approved and clinically tested agents (NIH Clinical Collection (Evotec, San Francisco, CA), LOPAC Pfizer (Sigma Aldrich), and the FDA-Approved Drug Library (Selleck Chemicals)) were tested at a single concentration of 10 µMin dimethyl sulfoxide (DMSO). We focused on inhibitor compounds that have single target genes (N = 237). Compounds were first diluted 1:50 in serum-free media, and then further diluted 1:20 into 96-well plates containing 5000 cells in 190 µl of keratinocyte SFM as described. Vehicle control (0.1% DMSO) was used as a negative control, and 1 µM staurosporine (Sigma Aldrich) was used as a positive cell lethality control in all plates. Each drug was tested for 72h under 21 and 0.1% $O_2$. The cells were fixed using 4% formaldehyde and stained with DAPI dilactate (Sigma Aldrich). We assess viability by quantifying the number of stained nuclei using the Operetta High-Content Imaging System (Perkin Elmer).

For gene knockout screening in liv7k cell line, transfections were performed in 384-well plates in 21% $O_2$ and 0.1% $O_2$. A complex of siRNA (Dharmacon, GE Healthcare) and 0.125 µL RNAiMax (Life Technologies) was added to each well and incubated for 30 min (final siRNA concentration 25 nM). 5000 cells were added per well in keratinocyte serum-free medium. The screen was conducted using siGenome RNAi pools and included All-Stars Cell Death control (Qiagen) and ON-TARGETplus Non-targeting Control Pool (Dharmacon, GE Healthcare). 72 h later, cells were fixed and stained with DAPI. The number of nuclei per well was counted using the Operetta High-Content Imaging System (Perkin Elmer).

For drug response screen in breast cancer cell lines, a panel of 7 breast cancer cell lines was screened using the drug repurposing libraries and workflow detailed for the liv7k cell line, under normoxic conditions only. All cell lines were maintained in Dulbecco's Modified Eagle Medium (DMEM) containing 10% FBS and 10% glutamine, with MCF10A cells receiving additional EGF (20 ng/ml), hydrocortisone (0.5 mg/ml), cholera toxin, (100 ng/ml) and insulin (10 µg/ml) supplementation. Cells were seeded at concentrations which resulted in approximately 80% well confluence after 72 h incubation (2000 > 8000).

For gene essentiality and drug response prediction, a gene was marked underactive if its expression level is less than 1/3-quantile of its level across all CCLE cell lines (N > 1000)[58] (see Supplementary Note 1). The growth inhibition induced by individual gene knock-outs was predicted by cSL-score of the gene, i.e., counting the number of inactive cSL partners in the given condition. The cSL network was inferred with FDR < 0.2 at the individual gene or drug level starting from the initial SL pool. We performed a large-scale gene knock-out screen to measure the growth inhibition effect of the genes over both the conditions

(Supplementary Data 6, Supplementary Note 1). To predict the conditional essentiality, we compared the cSL-score to hypoxic and normoxic conditions for the genes that are differential by essentiality and by cSL-score (Supplementary Note 1). The prediction accuracy was quantified by simple fold change comparison between our prediction and measured essentiality values.

For drug response prediction, we followed Procedure 1 (see "ISLE-based drug response prediction" in Methods above). We focused on 237 and 92 single-target drugs respectively for liv7k and BC cell lines. We considered %Growth Inhibition >top 33% as a positive set (responsive), and the rest as a negative set (non-responsive). We performed ROC and precision-recall analysis using the cSL-score as a prediction for drug response. We performed an analogous analysis to predict drug response in breast cancer cell lines. To predict the conditional essentiality in liv7k cell line, we compared the cSL-score of the drug targets and the drug's effectiveness in hypoxic and normoxic conditions (see Supplementary Note 1). The prediction accuracy was quantified by simple fold change comparison between our prediction and measured drug response values analogous to conditional essentiality analysis described above.

**Testing cSL-based drug synergy in patient-derived cell lines**. All DNA samples used in this study were derived from metastases. Samples used for whole-exome capture were extracted from cell lines established directly from patient tumors as described previously[76]. Briefly, a panel of pathology-confirmed metastatic melanoma tumor resections, paired with apheresis-collected peripheral blood mononuclear cells, were collected from patients enrolled in IRB-approved clinical trials at the Surgery Branch of the National Cancer Institute. Pathology-confirmed melanoma cell lines were derived from mechanically or enzymatically dispersed tumor cells, which were then cultured in RPMI 1640 + 10% FBS at 37 °C in 5% CO2 for 5–15 passages. Genomic DNA was isolated using DNeasy Blood & Tissue kit (Qiagen, Valencia, CA). For all samples, matching between germline and tumor DNA was verified by direct sequencing of 26 single nucleotide polymorphisms (SNP) at 24 loci. A subset of cell lines used in the study was derived from a panel of pathology-confirmed metastatic melanoma tumor resections collected from patients enrolled in institutional review board (IRB)-approved clinical trials at the Surgery Branch of the National Cancer Institute. Pathology-confirmed melanoma cell lines were derived from mechanically or enzymatically dispersed tumor cells, which were then cultured in RPMI-1640 supplemented with 10% FBS at 37 °C in 5% $CO_2$ for 5–15 passages. All cell lines tested negative for mycoplasma.

For proliferation assay, cells were seeded at a density of 4000 cells per well in 96-well plates. The next day, cells were treated with several concentrations and combinations of IGF1R inhibitor OSI906 (linsitinib, Selleckchem, 0.15–20 μM), Bcl-2 inhibitor ABT-263 (navitoclax, Adooq, 0.15-20 μM), or AKT inhibitor MK-2206 (Adooq, 0.15-20 μM) and PI3K inhibitor GDC-0941 (Adooq, 0.15-20 μM). After 72–96 h days, cell proliferation was assessed using the CellTiter-Glo reagent (Promega). IC50 graphs were determined using GraphPad Prism. All experiments were conducted in triplicates.

From the drug-cSL-network derived from pan-cancer screening, we first filtered the actionable targets relevant to melanoma, prioritizing the pathway alterations associated with the key melanoma drivers[63]. We then selected two among the top five pairs of the drug targets with the strongest ISLE-predicted cSL between them, quantified by cSL-pair-score (see Supplementary Note 1 for details) - AKT1 and PIK3CA, and BCL2L1 and IGF1R (Supplementary Data 9). Among the 29 patient-derived melanoma cell lines available to us, we selected for each gene pair the top three or four cell lines in which the target genes are highly expressed (Supplementary Data 10). Then we chose to continue our validation for the BCL2L1/IGF1R pair with three patient-derived cell lines (PDCs) and a melanoma cell line 501Mel and for the AKT1/PIK3CA pair with four PDCs, two of which overlaps with those used for BCL2L1/IGF1R. The synergy values were calculated according to the Chou-Talalay method[75], using the CompuSyn software (ComboSyn, Inc., Paramus, NJ). The drug interaction is quantified by Combination Index (CI) values, where CI < 1, CI = 1, and CI > 1 indicate synergistic, additive, and antagonistic effects, respectively.

**Predicting clinical drug response based on cSL**. We followed Procedure 1 (see 'ISLE-based drug response prediction' in Methods above) in predicting treatment outcome in patients, using transcriptomic data from tumor samples to determine the cSL-score, calculated based on the drug-cSL-network (Supplementary Fig. 8). To avoid any circularity in patients' drug response prediction, we excluded TCGA samples of patients who have treatment records (resulting in $N = 6268$) from the data used for SL identification (Step II). We evaluated our prediction vs. independent measures of drug response. In drug response prediction with TCGA samples, we have an explicit RECIST annotation regarding the pathological drug response, namely complete response (CR), partial response (PR), stable disease (SD), and progressive disease (PD) for a subset of patient samples. Based on this annotation, we divided the samples into responders (CR, PR, SD) vs. non-responders (PD) and compared their cSL-score using a Wilcoxon rank sum test. For each drug, we focused on highly specific drugs ($N_{target} \leq 3$) and the cancer types, which involves a sufficient number of samples ($N > 12$) per cancer type in TCGA (suggesting the drug is primarily used for the treatment of the cancer type) to eliminate noise. Vinka alkanoid drugs were removed from the analysis because

their mechanism of action is not compatible with SL. We confirmed that the significance is not achieved by randomly assigned cSL-score to the samples under consideration, and the empirical P-value is provided. The same approach was applied to distinguish the sensitive vs. resistant cases in ICGC ovarian cancer dataset.

For other datasets, we used patient survival as a surrogate for drug response, assuming that responders would have a better prognosis than non-responders. Our predictions were tested against patient survival data by examining if the patients that have high cSL-score show significantly better prognosis using Cox regression analysis while controlling for other confounders such as patient's age, cancer subtypes, and driver mutation, wherever they are available. Specifically, for taxane-anthracycline data, patient's age and breast cancer subtypes were controlled; and for erlotinib analysis, KRAS mutation was controlled. For erlotinib analysis, we identified top 76 SL partners of EGFR (based on cSL-pair-score) to compare with the 76-gene EMT signature of the original study[66] (see Supplementary Note 1).

**Code availability**. ISLE is implemented in R, using the SLURM distributed parallel computation infrastructure. The code and resulting cSL networks (genome-wise clinical-SL-network (Supplementary Fig. 1) and the above drug-cSL-network) are available on GitHub: https://github.com/jooslee/ISLE/. The networks can be explored using a freely available Cytoscape software[77].

**Data availability**. All relevant data are available in the Supplementary Data files, or from the corresponding author upon request.

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

## Acknowledgements

The authors thank Ze'ev Ronai, Max Leiserson, Allon Wagner for their helpful comments. J.S.L., N.A., W.R. and E.R. are partially supported by a grant from the Israeli Science Foundation (ISF) 41/11 and R33-CA225291-01. N.A. and W.R. are supported by the NCI-UMD partnership for integrative cancer research. Y.S. is supported by the ISF 696/17 and R.A. is supported by Clore foundation. S.H. is supported by NSF 1564785, and C.H.B. is supported by Welcome Trust award 102696. D.A., C.S., E.B. and T.G. are partially supported by Canadian friends Alex U. Soyka Foundation. M.D., L.M., D.J., and E.S. is supported by Cancer Research UK. L.J.-A. is supported by Cancer Research Institute Irvington Fellowship and Eric and Wendy Schmidt Postdoctoral program.

## Author contributions

J.S.L., A.D. and E.R. conceived and designed the research. J.S.L., A.D., E.S., Y.S., C.H.B., E.G., T.G., P.S.M., E.R. designed the experimental procedure. J.S.L. and A.D. performed the computational analysis and statistical computations. R.A., M.D., L.M., D.J., A.A., D.A., C.S., E.B., J.J.W., P.S.M., T.G., E.G., C.H.B., Y.S., E.S. performed the experiments. L.J.-A., N.A., S.G.P., G.S., K.C., W.R., S.H. helped with the computational analysis. J.S.L., A.

D., E.R. wrote the paper. J.S.L. and A.D. contributed equally to each other; L.J.-A. and R. A. contributed equally to each other; and Y.S. and E.S. contributed equally to each other.

## Additional information

**Competing interests:** The authors declare no competing interests.

