## [Peer Review File · Nature Communications]

Reviewers' comments:

Reviewer #3 (Remarks to the Author):

Lee et al have submitted a significantly revised manuscript with a different focus to the previous iterations. Rather than focusing on predicting new synthetic lethal interactions, the emphasis is now on prioritizing synthetic lethal interactions determined from experimental screens. The manuscript is considerably clearer and all of my previous concerns regarding overfitting etc have been addressed.

Specific comments :

The authors mention in the last line of the results and also in the discussion that they have also generated an miRNA/protein SL network. However this network is not used or discussed elsewhere in the manuscript so its inclusion is confusing.

'Under-expression' is not the same as a knockout and I think this tempers some of the discussion regarding the fraction of SLs supported by clinical data. If an SL is determined for a tumor suppressor loss of function mutation (e.g. in MSH2) and not observed in samples with functioning but low expressed MSH2 then I think it is difficult to infer much about the clinical support for samples with the loss of function mutation. Some discussion of the distinction between knockout and low expression would address this.

Section 4 (cSL-based prediction of drug response in patients) would benefit from an introduction to explain that the clinical predictions are made using the expression of synthetic lethal partners of the drugs used in each patient cohort.

Reviewer #4 (Remarks to the Author):

In the manuscript entitled "Harnessing synthetic lethality to predict the response to cancer treatment", the authors present a computational approach for inferring clinically relevant synthetic-lethal gene pairs from TCGA data; for this iteration of revisions, computational analysis was restricted to synthetic lethal genes and gene pairs from previously published experimental screens. The authors undertake an impressive and comprehensive validation of the presented clinically relevant synthetic lethal gene pairs, which utilized existing cancer pharmacogenomics datasets (e.g., CCLE, DREAM challenges, and mouse xenografts) and published patient response, and contributed new cell-based knock-out and drugging experiments. I believe this work will be of interest to the Nature Communications readership. Concerns include a somewhat non-intuitive and many-step approach, and use of non-standard FDR thresholds (e.g., $FDR \leq 0.1$ or $FDR \leq 0.2$), raising the concern of (potentially unintentional) P-hacking. Also, I found the paper difficult to read, and requiring substantial grammatical improvement and editing.

For this revision of the manuscript, I was solicited to review author responses to the critique leveled by a now-absentee Reviewer #1. The authors appear to have changed the paper substantially since

the previous submission, and Review #1's critiques are either addressed or no longer relevant; basing my decision solely on these changes, I believe the manuscript is acceptable for publication.

Reviewer #5 (Remarks to the Author):

Following the previous round of reviews, the authors have significantly revised their manuscript. Having gathered a large number of synthetic lethal interactions derived from recent studies, they now focus on determining which one of these are clinically relevant.

As requested by reviewers they now provide validation for ISLE-predicted SLi in five experimentally-identified gold standard SL datasets and compare ISLE with DAISY.

They've fixed the statistical issues identified by Reviewer 2.

Altogether I find these revisions satisfactory.

Common response to all reviewers

We appreciate the comprehensive review of our manuscript by all three reviewers. As requested, we have now updated the SL network using a more relaxed FDR threshold of 0.2 that is employed throughout the manuscript. We further clarified our analysis and methods as the reviewers suggested. All changes are marked in the text in a blue colored font.

Point-by-point response to reviewers

Response to Reviewer #3:

Lee et al have submitted a significantly revised manuscript with a different focus to the previous iterations. Rather than focusing on predicting new synthetic lethal interactions, the emphasis is now on prioritizing synthetic lethal interactions determined from experimental screens. The manuscript is considerably clearer and all of my previous concerns regarding overfitting etc have been addressed.

Specific comments :

The authors mention in the last line of the results and also in the discussion that they have also generated an miRNA/protein SL network. However this network is not used or discussed elsewhere in the manuscript so its inclusion is confusing.

Thanks. We have now removed any reference to the miRNA-protein SL network, keeping the paper focused on SL interactions involving protein encoding genes.

'Under-expression' is not the same as a knockout and I think this tempers some of the discussion regarding the fraction of SLs supported by clinical data. If an SL is determined for a tumor suppressor loss of function mutation (e.g. in MSH2) and not observed in samples with functioning but low expressed MSH2 then I think it is difficult to infer much about the clinical support for samples with the loss of function mutation. Some discussion of the distinction between knockout and low expression would address this.

Thanks for this insightful comment. We agree and as suggested, we now added a brief description of how the test was done in the main text (page 6), which reads:

“Importantly, to estimate the clinical relevance of these interactions, we used mutation data for isogenic screens; and for double knockdown screens, we used copy number and transcriptional data (see Supplementary Information Sec 1.1 for details). Furthermore, in addition to pan-cancer tests, we also

performed the clinical estimations outlined above in the specific cancer types of relevance, aiming to uncover the potential clinical relevance of the SLs screened in as comprehensive a manner as possible”

We further added a brief discussion of this issue in Discussion section (page 15), which reads:

“The clinical significance estimation of the initial pool of candidate SLs should be distinguished between two different cases: (1) First, in the case of the specific candidate SLs that arise from isogenic (or double-knockout) cell-lines screens, we specifically performed their clinical significance estimation by considering the corresponding mutation (or copy number and expression) data in the tumors and also by scanning the specifically relevant cancer types in which they have been inferred, so we believe that their clinical significance estimations are quite robust. (2) Second, the clinical significance estimations of the candidate SLs emerging from the single-knockdown screens are however probably less tight, as they are both tested and inferred via a pan-cancer analysis (Methods), while some of these candidate SLi may still be of clinical value in specific cancer types and contexts”

Section 4 (cSL-based prediction of drug response in patients) would benefit from an introduction to explain that the clinical predictions are made using the expression of synthetic lethal partners of the drugs used in each patient cohort.

Thanks. Such an explanation has been added and now reads (page 12):

“We next inferred the drug-cSL network of clinically approved cancer drugs (Methods, **Figure S8**) and tested its capacity to predict drug response in three different patients’ datasets of different tumor types⁶¹⁻⁶⁴. Following the procedure described before for *in vitro* drug prediction, for each drug we predict its response in each sample by computing its cSL-score; the cSL-score of a drug denotes the number of its target genes’ SL partners that are down-regulated in the specific sample divided by the number of the drug’s target genes (this number is determined from the sample’s transcriptomics data; Methods).”

Response to Reviewer #4:

In the manuscript entitled “Harnessing synthetic lethality to predict the response to cancer treatment”, the authors present a computational approach for inferring clinically relevant synthetic-lethal gene pairs from TCGA data; for this iteration of revisions, computational analysis was restricted to synthetic lethal genes and gene pairs from previously published experimental screens. The authors undertake an impressive and comprehensive validation of the presented clinically relevant synthetic lethal gene pairs, which utilized existing cancer pharmacogenomics datasets (e.g., CCLE, DREAM challenges, and mouse xenografts) and published patient response, and contributed new cell-based knock-out and drugging experiments. I believe this work will be of interest to the Nature Communications readership. Concerns include a somewhat non-intuitive and many-step approach, and use of non-standard FDR thresholds (e.g., $FDR \leq 0.1$ or $FDR \leq 0.2$), raising the concern of (potentially

unintentional) P-hacking. Also, I found the paper difficult to read, and requiring substantial grammatical improvement and editing.

Thanks for these very helpful comments. Indeed, in previous revision, we inferred the genome-wide SL network with FDR 0.1 but used an SL-network inferred with FDR 0.2 for essentiality and drug response predictions. To clarify, there was absolutely no P-hacking in our previous approach. Rather, it was motivated by our wish to have a relatively smaller-scale cancer SL-network that would be more amenable for visualization and presentation on one hand (hence the stricter FDR correction in that case) and, on the other hand, to obtain a larger amount of predicted SLs for specific drugs to boost the prediction signal (hence the use of the more relaxed threshold on that case). However, we understand the concern and after giving it more thought we decided to follow the reviewer's advice and revised the paper and are now consistently using an FDR threshold of 0.2 throughout the manuscript, as is now clearly noted in the first section of Methods (page 17). As the FDR= 0.2 is too large to visualize in an informative manner, we still present the smaller network (FDR=0.1) in the main text but give the full report of the larger resolution network in the form of Supplementary Figure (S1) and Data. In the process of standardizing the code, documenting it and preparing it for public dissemination, we noticed that a few analyses used slightly varied definitions of the SL-score function and we now have taken care that this is now completely uniform across the board too. The results remain qualitatively similar with fairly minute changes overall, except for one result (prediction of synergy in a melanoma drug response dataset) that has lost its significance and is hence now omitted. We also revised Figure 2 to include a specific comparison to top methods tested in the recent DREAM7 challenge, which we think is more interesting.

As for the multi-step approach, as each inference criteria on its own analyzed large scale biological data that is noisy to some extent, ISLE draws on combining a few such criteria and data types for determining cSLs, as summarized on page 15: “(1) First, the tumor data are noisy, both on the molecular and the survival side. To achieve a strong and robust predictive signal, we infer a common pan-cancer network by combining data from different tumor types, while obviously there is variability in SL_i at different cancer types.”

However, as we now explicate and note in the revised version, an SL is inferred only if it is supported by *all* selection criteria, and the specific order of different steps is chosen in a manner that minimizes the computing time, as now briefly described on page 4: “Conceptually, ISLE selects the clinically-relevant SL pairs that satisfy all of the three conditions outlined below. From a computational standpoint, however, it applies them in a specific sequential manner that minimizes the computational cost of their identification”

Finally, following the reviewer's suggestion we also make another comprehensive rewriting pass over all the paper aiming to further improve it grammatically and further improve its clarity (places where considerable changes were made are now marked in blue color throughout the text for your convenience).

For this revision of the manuscript, I was solicited to review author responses to the critique leveled by a now-absentee Reviewer #1. The authors appear to have changed the paper substantially since the previous submission, and Review #1's critiques are either addressed or no longer relevant; basing my decision solely on these changes, I believe the manuscript is acceptable for publication.

Thanks, and much appreciated.

Response to Reviewer #5:

Following the previous round of reviews, the authors have significantly revised their manuscript. Having gathered a large number of synthetic lethal interactions derived from recent studies, they now focus on determining which one of these are clinically relevant.

As requested by reviewers they now provide validation for ISLE-predicted SLi in five experimentally-identified gold standard SL datasets and compare ISLE with DAISY.

They've fixed the statistical issues identified by Reviewer 2.

Altogether I find these revisions satisfactory.

Thanks for your comments, and much appreciated.